# Diffusion Models for Open-Vocabulary Segmentation

## Abstract

The variety of objects in the real world is unlimited and is thus impossible to capture using models trained on a closed, pre-defined set of categories. Recently, open-vocabulary recognition has garnered significant attention, largely facilitated by advances in large-scale vision-language modelling. In this paper, we present OVDiff, a novel method that leverages the generative properties of text-to-image diffusion models for open-vocabulary segmentation. Specifically, we propose to synthesise support image sets from arbitrary textual categories, creating for each category a set of prototypes representative of both the category itself and its surrounding context (background). Our method relies solely on pre-trained components: segmentation is obtained by simply comparing a target image to the prototypes without further fine-tuning. We show that our method can be used to ground any pre-trained self-supervised feature extractor in natural language and provide explainable predictions by mapping back to regions in the support set. Our approach shows strong performance on a range of open-vocabulary segmentation benchmarks, obtaining a lead of more than 10% over prior work on PASCAL VOC.

## 1 Introduction

Semantic segmentation aims to classify each pixel in an image into a set of categorical labels. Traditionally, this task requires large, densely annotated datasets for training models to predict class assignments at pixel level and relies on the assumption that the set of labels is fixed and predefined. Collecting and annotating such data is not only cumbersome and costly, but it also results in static models that are difficult to extend to new categories.

Open-vocabulary semantic segmentation relaxes this restriction by allowing nearly arbitrary free-form text queries as class descriptions. This problem is often approached by extracting image embeddings and matching them to a representation of the text queries. Obtaining these embeddings is challenging as they need to describe the image densely and they must also be compatible with the representation of any possible text query. Prior work addresses this challenge by starting from multi-modal representations (*e.g.*, CLIP (Radford et al., 2021)) to bridge vision and language and further relies on labelled data to fine-tune the representations for the segmentation task. Hence, in line with the zero-shot setting (Bucher et al., 2019), these methods require dense annotations for some known categories, while also extending segmentation to unseen categories by incorporating language.

An alternative, which eliminates the need for collecting ad-hoc manual annotations, is to leverage image-text pairs that can be obtained at scale by crawling the Internet. Existing methods (Xu et al., 2022a; Ren et al., 2023; Xu et al., 2023b; Luo et al., 2022; Mukhoti et al., 2022; Cha et al., 2022) observe that large-scale vision-language models such as CLIP have a limited understanding of the positioning of objects within an image and extend these models with additional grouping mechanisms for better localisation using only image-level captions, but no mask supervision. This, however, requires additional contrastive training at scale. Despite yielding promising results, there are some pitfalls to this approach. Firstly, as text might not describe all entities in the image or might mention elements that are not depicted, the training is noisy. Secondly, similar captions may be used to describe a wide range of visual appearances or a similar concept might be described in different ways, though image and language are processed independently. Lastly, most methods resort to heuristics to segment the background (*i.e.*, leave some pixels unlabelled), as it often cannot be described as a textual category. The usual approach is to threshold the similarities to all categories. Finding an

appropriate threshold, however, can be challenging and may vary depending on the image, often resulting in imprecise object boundaries. Effectively handling the background remains an open issue.

While the field of Computer Vision evolves towards large, pre-trained, general-purpose models, its applications still rely on task-specific approaches, data, and fine-tuning. Thus, in this work, we show that the segmentation problem can be effectively tackled with a combination of frozen "foundation" models without any task-specific adaptation.

Specifically, we show that large-scale text-to-image generative models such as StableDiffusion (Rombach et al., 2022) open up new avenues for solving this problem, as they are able to bridge the vision-language gap by synthesising data on-the-fly, but also produce latent spaces that are semantically meaningful and well-localised. This also solves a second problem: multi-modal embeddings are difficult to learn and often suffer from ambiguities and differences in detail between modalities. Instead, our approach can use unimodal features for open-vocabulary segmentation, which offers several advantages. Firstly, as text-to-image generators encode a distribution of possible images, this offers a means to deal with intra-class variation and captures the ambiguity in textual descriptions. Secondly, the generative image models encode not only the visual appearance of objects but also provide contextual priors such as backgrounds which can greatly improve the segmentation quality.

Given a textual prompt, our method, OVDiff, uses a generative model to produce a support set of visual examples that we then decompose into a set of feature prototypes at different levels of granularity: class, instance, and part prototypes. Prototypes are essentially image features extracted from off-the-shelf unsupervised feature extractors. They can then be used in a simple nearest-neighbour lookup scheme to segment any image. We also propose to leverage the backgrounds from sampled images to encode a set of negative prototypes that enable direct background segmentation.

In this work, we present a simple framework that achieves state-of-the-art performance across open-vocabulary segmentation benchmarks. It makes use of several off-the-shelf pre-trained networks and requires no additional data nor fine-tuning. As such, the model can directly benefit from future improvements of its adopted models.

## 2 RELATED WORK

**Zero-shot open-vocabulary segmentation.** Open-vocabulary semantic segmentation is a relatively new problem and is typically approached in two different ways. The first line of work poses the problem as a "zero-shot" task, *i.e.*, segmenting unseen classes after training on a set of observed classes with dense annotations. Early approaches (Bucher et al., 2019; Li et al., 2020; Gu et al., 2020; Cheng et al., 2021) explore generative networks to sample features using conditional language embeddings for classes. In (Xian et al., 2019; Li et al., 2021) image encoders are trained to output dense features that can be correlated with word2vec (Mikolov et al., 2013) and CLIP (Radford et al., 2021) text embeddings. Follow-up works (Ghiasi et al., 2022; Liang et al., 2022; Ding et al., 2022; Xu et al., 2022b) approach the problem in two steps, predicting class-agnostic masks and aligning the embeddings of these masks with language. IFSeg (Yun et al., 2023) generates synthetic feature maps by pasting CLIP text embeddings into a known spatial configuration to use as additional supervision. Different from our approach, all these works rely on mask supervision for a set of known classes.

The second line of work eliminates the need for mask annotations and instead aims to align image regions with language using only image-text pairs. This is largely enabled by recent advancements in large-scale vision-language models (Radford et al., 2021). Some methods introduce internal grouping mechanisms such as hierarchical grouping (Xu et al., 2022a; Ren et al., 2023), slot-attention (Xu et al., 2023b), or cross-attention to learn cluster centroids (Liu et al., 2022; Luo et al., 2022). Assignment to language queries is performed at group level. An alternative line of work (Zhou et al., 2022; Mukhoti et al., 2022; Cha et al., 2022; Ranasinghe et al., 2022) aims to learn dense features that are better localised when correlated with language embeddings at pixel level. With the exception of (Ranasinghe et al., 2022; Zhou et al., 2022), thresholding is often required to determine the background during inference. Alternatively, Ranasinghe et al. (2022) use a curated list of background prompts.

Our method falls into the second category. However, in contrast to prior work, we leverage a generative model to translate language queries to pre-trained image feature extractors without further training. We also segment the background directly, without relying on thresholding or curated list of background prompts. A closely related approach to ours is ReCO (Shin et al., 2022b), where CLIP is

used for image retrieval compiling a set of exemplar images from ImageNet for a given language query, which is then used for co-segmentation. In our method, the shortcoming of an image database is addressed by synthesising data on-demand. Furthermore, instead of co-segmentation, we leverage the cross-attention of the generator to extract objects. Instead of similarity of support images, our method leverages diverse samples and makes use of both foreground and contextual backgrounds.

**Diffusion models.** Diffusion models (Sohl-Dickstein et al., 2015; Ho et al., 2020; Song et al., 2021) are a class of generative methods that have seen tremendous success in text-to-image systems such as DALL-E (Ramesh et al., 2022), Imagen (Saharia et al., 2022), and Stable Diffusion (Rombach et al., 2022), trained on Internet-scale data such as LAION-5B (Schuhmann et al., 2022). The step-wise generative process and the language conditioning make pre-trained diffusion models attractive also for discriminative tasks. They have been recently used in few-shot classification (Zhang et al., 2023), few-shot segmentation (Baranchuk et al., 2022) and panoptic segmentation (Xu et al., 2023a), and to generate pairs of images and segmentation masks (Li et al., 2023b). However, these methods rely on dense manual annotations to associate diffusion features with the desired output.

Annotation-free discriminative approaches such as (Li et al., 2023a; Clark & Jaini, 2023) use pre-trained diffusion models as zero-shot classifiers. DiffuMask (Wu et al., 2023) uses prompt engineering to synthesise a dataset of "known" and "unseen" categories and trains a closed-set segmenter with masks obtained from the cross-attention maps of the diffusion model. DiffusionSeg (Ma et al., 2023) uses DDIM inversion (Song et al., 2021) to obtain feature maps and attention masks of object-centric images to perform unsupervised object discovery, but relies on ImageNet labels and is not open-vocabulary. Our approach also leverages the rich semantic information present in diffusion models for segmentation; unlike these methods, however, it is open-set and does not require further training.

**Unsupervised segmentation.** Our work is also related to unsupervised segmentation approaches. While early works relied on hand-crafted priors (Cheng et al., 2015; Wei et al., 2012; Zhang et al., 2018; Zeng et al., 2019; Nguyen et al., 2019) later approaches leverage feature extractors such as DINO (Caron et al., 2021) and perform further analysis of these methods (Wang et al., 2022b; Melas-Kyriazi et al., 2022a; Siméoni et al., 2021; Siméoni et al., 2022; Hamilton et al., 2022; Shin et al., 2022a; Wang et al., 2023; 2022a). Some approaches make use of generative methods, usually GANs, to separate images in foreground and background layers (Bielski & Favaro, 2019; Chen et al., 2019; Benny & Wolf, 2020; Bielski & Favaro, 2022) or analyse latent structure to induce known foreground-background changes (Voynov et al., 2021; Melas-Kyriazi et al., 2022b) to synthesise a training dataset with labels. Largely focused on unsupervised saliency prediction, these methods are class-agnostic and do not incorporate language.

## 3 METHOD

We present OVDiff, a method for open-vocabulary segmentation, *i.e.*, semantic segmentation of any category described in natural language. To achieve this goal we (1) leverage *text-to-image generative models* to generate a set of images representative of the described category, and (2) use these to ground off-the-shelf *pretrained feature extractors*. This process does not require further training: it relies only on pretrained components and does not use additional training data or parameter finetuning.

Our goal is to devise an algorithm which, given a new vocabulary of categories $c_i \in \mathcal{C}$ formulated as natural language queries, can segment any image against it. Let $I \in \mathbb{R}^{H \times W \times 3}$ be an image to be segmented. Let $\Phi_v : \mathbb{R}^{H \times W \times 3} \mapsto \mathbb{R}^{H' W' \times D}$ be an off-the-shelf visual feature extractor and $\Phi_t : \mathbb{R}^{d_t} \mapsto \mathbb{R}^D$ a text encoder. Assuming that image and text encoders are aligned, one can achieve zero-shot segmentation by simply computing a similarity function, for example, the cosine similarity $s(\Phi_v(I), \Phi_t(c_i))$, with $s(x, y) = \frac{x^T y}{\|x\|\|y\|}$, between the encoded image $\Phi_v(I)$ and an encoding of a class label $c_i$, which is a simple extension of the zero-shot classification paradigm to dense visual representations. To meaningfully compare different modalities, image and text features must lie in a shared representation space, which is typically learned by jointly training $\Phi_v$ and $\Phi_t$ using image-text or image-label pairs (Radford et al., 2021).

We propose two modifications to this approach. First, we observe that it is better to compare representations of the *same* modality than across vision and language modalities. We thus replace $\Phi_t(c_i)$ with a $D$-dimensional *visual* representation $\bar{P}$ of class $c_i$, which we refer to as a *prototype*.

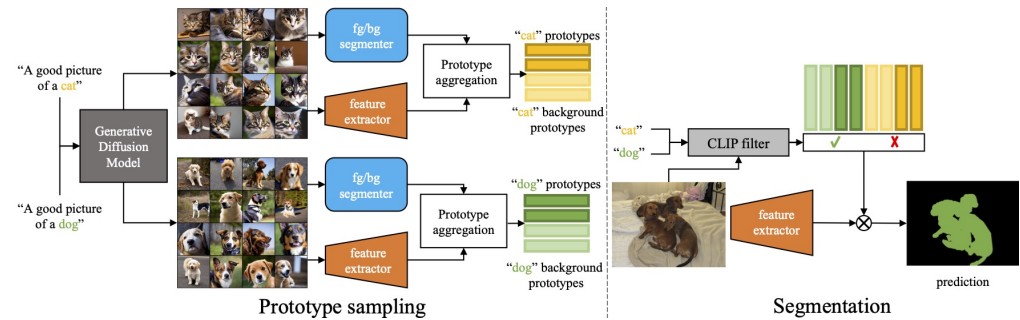

Figure 1: OVDiff overview. Prototype sampling: text queries are used to sample a set of support images which are further processed by a feature extractor and a segmenter forming positive and negative (background) prototypes. Segmentation: image features are compared against prototypes.The CLIP filter removes irrelevant prototypes based on global image contents.

In this case, the same feature extractor can be used for both prototypes and target images, thus their comparison becomes straightforward and does not necessitate further training.

Second, we propose utilizing *multiple* prototypes per category instead of a single class embedding. This enables us to accommodate intra-class variations in appearance and, as we explain later, it also allows us to exploit contextual priors, which in turn help to effectively segment the background. Finally, our approach handles the queries $c_i$ independently, allowing for arbitrary changes to the target vocabulary $\mathcal{C}$ without the need for recomputation.

**Support set generation.** To construct a set of prototypes in the visual domain, the first step of our approach is to sample a support set of images representative of each category $c_i$. This can be accomplished by leveraging pretrained text-conditional generative models. Sampling images from a generative model, as opposed to a curated dataset of real images, aligns well with the goals of open-vocabulary segmentation as it enables the construction of prototypes for *any* user-specified category or description, even those for which a manually labelled set may not be readily available (*e.g.*, $c_i = $ "donut with chocolate glaze").

Specifically, for each query $c_i$, we define a prompt "A good picture of a $\langle c_i \rangle$" and generate a small batch of $N$ support images $\mathcal{S} = \{S_1, S_2, \ldots, S_N \mid S_n \in \mathbb{R}^{hw \times 3}\}$ of height $h$ and width $w$ using Stable Diffusion (Rombach et al., 2022). In its most naïve form, a prototype $\bar{P}$ could then be constructed by averaging all features across all images. However, this is unlikely to result in a good prototype, because not all pixels in the generated image correspond to the category specified by $c_i$. To address this issue, we propose to extract the class prototypes as follows.

**Class prototypes.** Our approach generates two sets of prototypes, positive and negative, for each class. Positive prototypes are extracted from image regions that are associated with $\langle c_i \rangle$, while negative prototypes represent "background" regions. While considering negative or "background" prototypes is not strictly necessary for segmentation, we found these help to disambiguate objects from their surroundings by considering contextual priors, which greatly improves performance.

Thus, to obtain prototypes, the first step is segmenting the sampled images into foreground (representing $c_i$) and background regions. To identify regions most associated with $c_i$, we use the fact that the layout of a generated image is largely dependent on the cross-attention maps of the diffusion model (Hertz et al., 2022), *i.e.*, pixels attend more strongly to words that describe them. For a given word or description (in our case $c_i$), one can generate a set of attribution maps $\mathcal{A} = \{A_1, A_2, \ldots, A_N \mid A_n \in \mathbb{R}^{hw}\}$, corresponding to the support set $\mathcal{S}$, by summing the cross-attention maps across all layers, heads, and denoising steps of the network (Tang et al., 2022). Yet, thresholding these attribution maps may not be optimal for foreground/background segmentation, as they are often coarse or incomplete, and sometimes only parts of objects receive high activation.

To address this issue and ensure higher quality masks, we propose to use an unsupervised instance segmentation method, such as CutLER (Wang et al., 2023). This approach does not use prompts for object selection and may result in multiple binary object proposals. We denote these as $\mathcal{M}_n = \{M_{nr} \mid M_{nr} \in \{0, 1\}^{hw}\}$, where $n$ indexes the support images and $r$ indexes the object masks

(including a mask for the background). We thus introduce a promptable extension of CutLER: for each image, we select from $\mathcal{M}_n$ the mask with the highest (lowest) average attribution as the foreground (background):

$$M_n^{\mathrm{fg}} = \underset{M \in \mathcal{M}_n}{\arg\max} \frac{M^\top A_n}{M^\top M}, \quad M_n^{\mathrm{bg}} = \underset{M \in \mathcal{M}_n}{\arg\min} \frac{M^\top A_n}{M^\top M}. \tag{1}$$

We can then compute prototypes $P_n^{\mathrm{g}}$ for foreground and background regions ($\mathrm{g} \in \{\mathrm{fg}, \mathrm{bg}\}$) as

$$P_n^{\mathrm{g}} = \frac{(\hat{M}_n^{\mathrm{g}})^\top \Phi_v(S_n)}{m_n^{\mathrm{g}}} \in \mathbb{R}^D, \tag{2}$$

where $\hat{M}_n^{\mathrm{g}}$ denotes a resized version of $M_n^{\mathrm{g}}$ that matches the spatial dimensions of $\Phi_v(S_n)$, and $m_n^{\mathrm{g}} = (\hat{M}_n^{\mathrm{g}})^\top \hat{M}_n^{\mathrm{g}}$ counts the number of pixels within each mask. In other words, prototypes are obtained by means of an off-the-shelf pretrained feature extractor and computed as the average feature within each mask. We refer to these as *instance-level* prototypes, because they are computed from each image individually and each image in the support set can be viewed as an instance of class $c_i$.

In addition to instance prototypes, we found it helpful to also compute *class-level* prototypes $\bar{P}^{\mathrm{g}}$ by averaging the instance prototypes weighted by their mask sizes as $\bar{P}^{\mathrm{g}} = \sum_{n=1}^{N} m_n^{\mathrm{g}} P_n^{\mathrm{g}} / \sum_{n=1}^{N} m_n^{\mathrm{g}}$.

Finally, we propose to augment the set of class and instance prototypes using $K$-Means clustering of the masked features to obtain *part-level* prototypes. We perform clustering separately on foreground and background regions and take each cluster centroid as a prototype $P_k^{\mathrm{g}}$ with $1 \leq k \leq K$. The intuition behind this is to enable segmentation at the level of parts, support greater intra-class variability, and a wider range of feature extractors that might not be scale invariant.

We consider the union of all these feature prototypes

$$\mathcal{P}^{\mathrm{g}} = \bar{P}^{\mathrm{g}} \cup \{P_n^{\mathrm{g}} \mid 1 \leq n \leq N\} \cup \{P_k^{\mathrm{g}} \mid 1 \leq k \leq K\}, \mathrm{g} \in \{\mathrm{fg}, \mathrm{bg}\} \tag{3}$$

and associate all of them with a single category. We note that this process is repeated for each $c_i \in \mathcal{C}$ and we thus refer to $\mathcal{P}^{\mathrm{fg}}$ (and $\mathcal{P}^{\mathrm{bg}}$) as $\mathcal{P}_{c_i}^{\mathrm{fg}}$ ($\mathcal{P}_{c_i}^{\mathrm{bg}}$), *i.e.*, as the foreground (background) prototypes of class $c_i$. Since $\mathcal{P}_{c_i}^{\mathrm{fg}}$ ($\mathcal{P}_{c_i}^{\mathrm{bg}}$) depend only on class $c_i$, they can be precomputed, and the set of classes can be dynamically expanded without the need to adapt existing prototypes.

**Open-vocabulary segmentation.** To perform segmentation of any target image $I$ given a vocabulary $\mathcal{C}$, we first extract image features using the same visual encoder $\Phi_v$ used for the prototypes. The vocabulary is expanded with an additional background class $\hat{\mathcal{C}} = \{c_{\mathrm{bg}}\} \cup \mathcal{C}$, for which the positive (*foreground*) prototype is the union of all *background* prototypes in the vocabulary: $\mathcal{P}_{c_{\mathrm{bg}}}^{\mathrm{fg}} = \bigcup_{c_i \in \mathcal{C}} \mathcal{P}_{c_i}^{\mathrm{bg}}$. Then, a segmentation map can simply be obtained by comparing dense image features to prototypes using cosine similarity. A class with the highest similarity in its prototype set is chosen:

$$M = \underset{c \in \hat{\mathcal{C}}}{\arg\max} \max_{P \in \mathcal{P}_c^{\mathrm{fg}}} s(\Phi_v(I), P). \tag{4}$$

**Category pre-filtering.** To limit the impact of spurious correlations that might exist in the feature space of the visual encoder, we introduce a pre-filtering process for the target vocabulary given image $I$. Specifically, we leverage CLIP (Radford et al., 2021) as a strong open-vocabulary classifier but propose to apply it in a multi-label fashion to constrain the segmentation to the subset of categories $\mathcal{C}' \subseteq \mathcal{C}$ that appear in the target image. First, we encode the target image and each category using CLIP. Any categories that do not score higher than $1/|\mathcal{C}|$ are removed from consideration, that is we keep the subset $\{P_{c'}^{\mathrm{g}} \mid c' \in \mathcal{C}'\}, \mathrm{g} \in \{\mathrm{fg}, \mathrm{bg}\}$. If more than $\eta$ categories are present, then the top-$\eta$ are selected. We then form "multi-label" prompts as "$\langle c_a \rangle$ and $\langle c_b \rangle$ and $\ldots$" where the categories are selected among the top scoring ones taking into account all $2^\eta$ combinations. The best-scoring multi-label prompt determines the final list of categories to be used in Equation (4).

**"Stuff" filtering.** Occasionally, $c_i$ might not describe a countable object category but an identifiable region in the image, *e.g.*, sky, often referred to as a "stuff" class. "Stuff" classes warrant additional consideration as they might appear as background in images of other categories, *e.g.*, boat images might often contain regions of water and sky. As a result, the process outlined above might sample

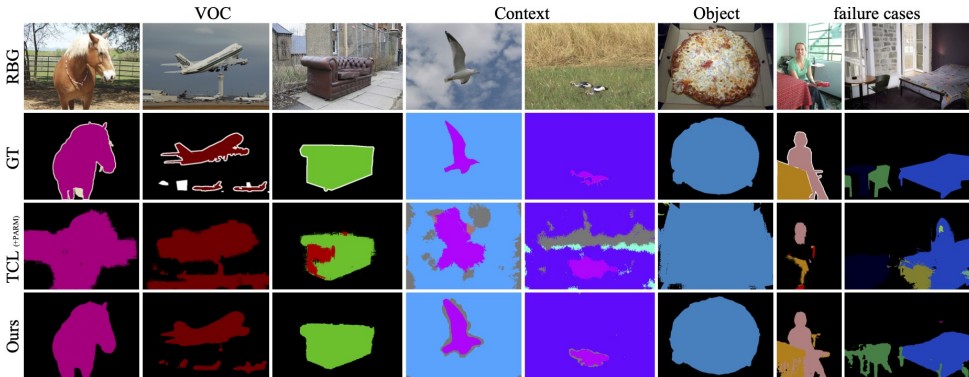

Figure 2: Qualitative results. OVDiff in comparison to TCL (+ PAMR). OVDiff provides more accurate segmentations across a range objects and stuff classes with well defined object boundaries that separate from the background well. Last 2 columns show failure cases. Additional table that appears in the background is segmented. Bed frame legs get misclassified as chairs.

background prototypes for one class that coincide with the foreground prototypes of another. To mitigate this issue, we introduce an additional filtering step to detect and reject such prototypes, when the full vocabulary, *i.e.*, the set of classes under consideration, is known. First, we only consider foreground prototypes for "stuff" classes. Additionally, any negative prototypes of "thing" classes with high cosine similarity with any of the "stuff" class prototypes are simply removed. In our experiments, we use ChatGPT (OpenAI, 2023) to automatically categorise a set of classes as "thing" or "stuff". While this categorisation may contain some errors, this filtering step is still beneficial.

## 4 EXPERIMENTS

We evaluate OVDiff on the open-vocabulary semantic segmentation task. First, we consider different feature extractors and investigate how they can be grounded by leveraging our approach. We then turn to comparisons of our method with prior work. We ablate the components of OVDiff, visualize the prototypes, and conclude with a qualitative comparison with prior works on in-the-wild images.

**Datasets and implementation details.** As the approach does note require further training of components, we only consider data used for evaluation. Following prior work (Xu et al., 2022a), to assess the segmentation performance, we report mean Intersection-over-Union (mIoU) on validation splits of PASCAL VOC (VOC) (Everingham et al., 2012), PASCAL Context (Context) (Mottaghi et al., 2014) and COCO-Object (Object) (Caesar et al., 2018) datasets, with 20, 59, and 80 foreground classes, respectively. All datasets have a background class as well. Context also contains both "things" and "stuff" classes. Similarly to Cha et al. (2022), we employ a sliding window approach. We use two scales to aid with the limited resolution of off-the-shelf feature extractors with square window sizes of 448 and 336, and a stride of 224 pixels. We set the size of the support set to $N = 32$. We detail further specifications of the sampling and other hyper-parameters in Appendix B.5.

### 4.1 GROUNDING FEATURE EXTRACTORS

Our method can be used in combination with *any* pretrained visual feature extractor for constructing prototypes and extracting image features. To verify this quantitatively, we experiment with various self-supervised ViT feature extractors (Table 2): DINO (Caron et al., 2021), MAE (He et al., 2022), and CLIP (Radford et al., 2021). We also experiment with SD as a feature extractor. We provide feature extraction details in Appendix B.2.

We find that SD performs the best, though CLIP and DINO also show strong performance based on our experiments on VOC. MAE shows the weakest performance, which may be attributed to its lack of semanticity (He et al., 2022); yet it is still competitive with the majority of purposefully trained networks when employed as part of our approach. We find that taking *keys* of the second to last layer in CLIP yields better results than using patch tokens (CLIP token). As feature extractors have

Table 1: Open-vocabulary segmentation. Comparison of our approach to the state of the art (under the mIoU metric). Our results are an average of 5 seeds $\pm\sigma$. *results from (Cha et al., 2022).

| Method | Support Set | Further Training | VOC | Context | Object |
|---|---|---|---|---|---|
| ReCo* (Shin et al., 2022b) | Real | ✗ | 25.1 | 19.9 | 15.7 |
| ViL-Seg (Liu et al., 2022) | ✗ | ✓ | 37.3 | 18.9 | - |
| MaskCLIP* (Zhou et al., 2022) | ✗ | ✗ | 38.8 | 23.6 | 20.6 |
| TCL (Cha et al., 2022) | ✗ | ✓ | 51.2 | 24.3 | 30.4 |
| CLIPpy (Ranasinghe et al., 2022) | ✗ | ✓ | 52.2 | - | 32.0 |
| GroupViT (Xu et al., 2022a) | ✗ | ✓ | 52.3 | 22.4 | - |
| ViewCo (Ren et al., 2023) | ✗ | ✓ | 52.4 | 23.0 | 23.5 |
| SegCLIP (Luo et al., 2022) | ✗ | ✓ | 52.6 | 24.7 | 26.5 |
| OVSegmentor (Xu et al., 2023b) | ✗ | ✓ | 53.8 | 20.4 | 25.1 |
| **OVDiff (Ours)** | Synthetic | ✗ | **66.3 ± 0.2** | **29.7 ± 0.3** | **34.6 ± 0.3** |
| TCL (Cha et al., 2022) (+ PAMR) | ✗ | ✓ | 55.0 | 30.4 | 31.6 |
| **OVDiff (+ PAMR)** | Synthetic | ✗ | **68.4 ± 0.2** | **31.2 ± 0.4** | **36.2 ± 0.4** |

Table 2: Segmentation performance of OVDiff based on different feature extractors.

| Feature Extractor | VOC |
|---|---|
| MAE | 54.9 |
| DINO | 59.1 |
| CLIP (token) | 51.4 |
| CLIP (keys) | 61.8 |
| SD | 64.4 |
| SD + DINO + CLIP | 66.4 |

Table 3: Ablation of different components. Each component is removed in isolation, measuring the drop ($\Delta$) in mIoU on VOC and Context datasets. Using SD features.

| Configuration | VOC | $\Delta$ | Context | $\Delta$ |
|---|---|---|---|---|
| Full | 64.4 | | 29.4 | |
| w/o bg prototypes | 53.2 | -11.2 | 28.9 | -0.5 |
| w/o category filter | 54.4 | -10.0 | 25.2 | -4.2 |
| w/o "stuff" filter | n/a | | 26.9 | -2.5 |
| w/o CutLER | 60.4 | -4.0 | 27.6 | -1.8 |
| w/o sliding window | 62.2 | -2.2 | 28.6 | -0.8 |
| only average $\bar{P}$ | 62.5 | -1.9 | 28.4 | -1.0 |

different training objectives, we hypothesise that their feature spaces might be complementary, thus we also consider an ensemble approach. In this case, the cosine distances formed between features of different extractors and respective prototypes are simply averaged. The combination of SD, DINO, and CLIP performs the best. We adopt this formulation for the main set of experiments.

## 4.2 COMPARISON TO EXISTING METHODS

In Table 1, we compare our method with prior work on three datasets: VOC, Context, Object. We include brief overview of the methods in Appendix B.4. We find that our method compares favourably, outperforming other methods in all settings. In particular, results on VOC show the largest margin, with more than 10% improvement over prior work. We hypothesise that this setting is particularly favourable to our method as it contains scenes where classes take up larger areas of the image.

In the same table, we also combine our method with PAMR (Araslanov & Roth, 2020), the post-processing approach employed by TCL. We find that it improves results for our method though improvements are less drastic since our method already yields better segmentation and boundaries.

Qualitative results are shown in Fig. 2. This figure highlights a key benefit of our approach: the ability to exploit contextual priors through the use of background prototypes, which in turn allows for the directly assignment of pixels to a background class. This improves segmentation quality because it makes it easier to differentiate objects from the background and to delineate their boundaries. In comparison, TCL predicts very coarse semantic masks and a larger amount of noise.

## 4.3 ABLATIONS

Next, we ablate the components of OVDiff on VOC and Context datasets. For these experiments, only SD is employed as a feature extractor. We remove individual components and measure the

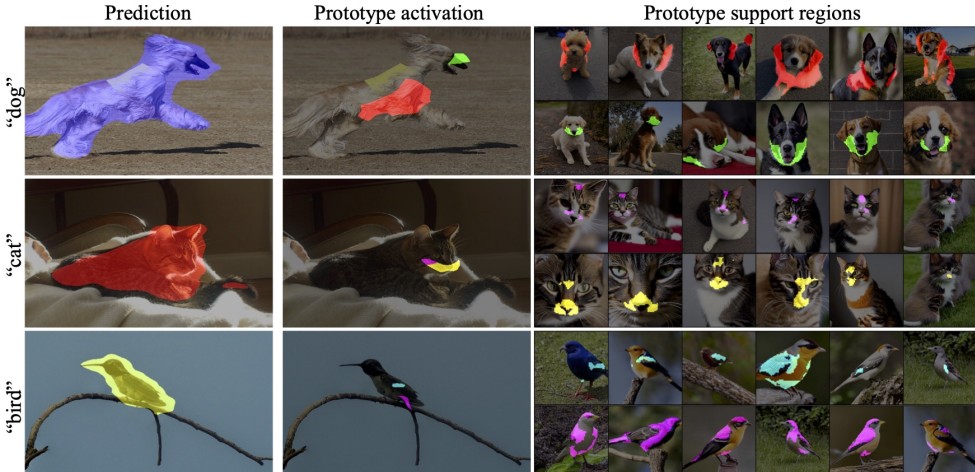

Figure 3: Analysis of the segmentation output by linking regions to samples in the support set. Left: our results for different classes. Middle: select color-coded regions "activated" by different prototypes for the class. Right: regions in the support set images corresponding to these (part-level) prototypes.

change in segmentation performance, summarising the results in Table 3. Our first observation is that background prototypes have a major impact on performance. When removing them from consideration, we instead threshold the similarity scores of the images with the foreground prototypes (set to 0.72, determined via grid search); in this case, the performance drops significantly, which again highlights the importance of leveraging contextual priors. On Context, the impact is less significant, likely due to the fact that the dataset contains "stuff" categories. Removing the *instance-* and *part-level* prototypes also negatively affects performance. Additionally, removing the category pre-filtering has a major impact. We hypothesize that this introduces spurious correlations between prototypes of different classes. On Context, "stuff" filtering is also important. Next, we evaluate the importance of using CutLER to obtain foreground/background prototypes.

Instead of a segmentation method, one can threshold the attribution maps obtained directly through the diffusion process. However, we find that this slightly reduces performance. Overall, background prototypes and pre-filtering contribute the most.

Finally, we measure the effect of varying the size of the support set $N$ in Fig. 4. We find that even at a low number of samples for each query, our method already shows strong performance. With increasing the number of samples, the performance improves, saturating at around 32, which is what we use in our main experiments. We leave additional ablations for Appendix C.2.

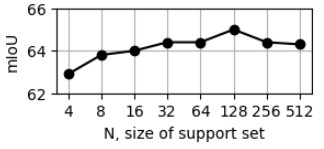

Figure 4: PascalVOC results with increasing support size $N$.

## 4.4 EXPLAINING SEGMENTATIONS

We inspect how our method segments certain regions by considering which prototype from $\mathcal{P}_c^{\mathrm{fg}}$ was used to assign a class $c$ to a pixel. Prototypes have a mapping to regions in the support set from where they were aggregated, *e.g.*, instances prototypes are associated with foreground masks $M_n^{\mathrm{fg}}$ and part prototypes with centroids/clusters.

By following these mappings, a set of support image regions can be retrieved for each segmentation decision providing a degree of explainability. Fig. 3 illustrates this for examples of dog, cat, and bird classes. For visualisation purposes, select prototypes and corresponding regions are shown. On the left, we show the full segmentation result of each image. In the middle, we select regions that correlated best with certain prototypes of the class. On the right, we retrieve images from the support set and highlight where each prototype emerged. We find that meaningful part segmentation merges due to clustering the support image features, and similar regions are segmented by corresponding prototypes. Though sometimes region covered in the input image will not fully align with whole

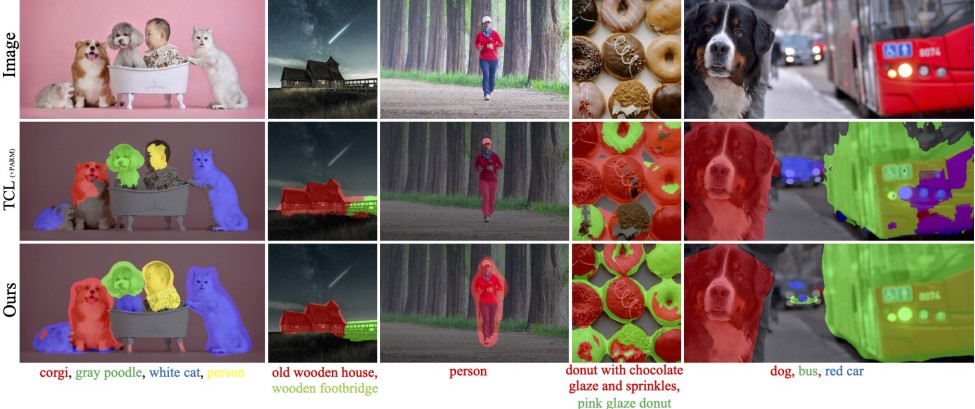

Figure 5: Qualitative comparison on in-the-wild images with TCL, which struggles with object boundaries, missing parts of objects, or including surroundings. Our method has more appropriate boundaries but does produce small halo effect around objects due to upscaling of feature extractors.

prototype (*e.g.* `cat`'s face around the eyes or lower belly/tail of `bird`). This shows how each segmentation produced by OVDiff is explained by precise regions in a small set of support images.

### 4.5 IN-THE-WILD

In Fig. 5, we investigate OVDiff on in-the-wild images containing simple and complex backgrounds. We compare with TCL+PAMR. In the first three images, both methods correctly detect the objects identified by the queries. TCL however misses parts of the objects, such as most of the person, and parts of animal bodies. The distinction between the house and the bridge in the second image is also better with OVDiff. We note that our segmentations sometimes have halos around objects. This is caused by the upscaling of the low-resolution feature extractor (SD in this case). The last two images contain difficult scenarios where both approaches struggle. The fourth image only contains similar objects of the same type. Both methods incorrectly identify plain donuts as either of the specified queries. OVDiff however correctly identifies chocolate donuts with varied sprinkles and separates all donuts from the background. In the final picture, the query "red car" is added, although no such object is present. The extra query causes TCL to incorrectly identify parts of the red bus as a car. Both methods incorrectly segment the gray car in the distance. However, overall, our method is more robust and delineates objects better despite the lack of specialized training or post-processing.

### 4.6 LIMITATIONS

As OVDiff relies on pretrained components, it inherits some of their limitations. OVDiff works with the limited resolution of feature extractors, due to which it might miss tiny objects. While this can be partially mitigated with a sliding window, employing high-resolution feature extractors is one direction of future improvements. Furthermore, OVDiff cannot segment what the generator cannot generate. For example, current diffusion models struggle with producing legible text. Finally, one limitation comes from the computational overhead of sampling support images. We observe that in practice often whole image collections are segmented by the same set of queries, amortising this cost.

### 5 CONCLUSION

We introduce OVDiff, an open-vocabulary segmentation method that operates in two stages. First, given queries, support images are sampled and their features are extracted to create class prototypes. These prototypes are then compared to features from an inference image. This approach offers multiple advantages without task-specific adaptation of its pre-trained components: diverse prototypes accommodating various visual appearances and negative prototypes for background localisation. OVDiff outperforms prior work on benchmarks, exhibiting fewer errors, effectively separating objects from background, and providing explainability through segmentation mapping to support set regions.

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

# SUPPLEMENTARY MATERIAL

In this supplementary material, we consider the broader impacts of our work (Appendix A), provide additional details concerning the implementation (Appendix B), and conclude with additional results (Appendix C).

## A    BROADER IMPACT

Semantic segmentation is a component in a very large and diverse spectrum of applications in healthcare, image processing, computer graphics, surveillance and more. As for any foundational technology, applications can be good or bad. OVDiff is similarly widely applicable. It also makes it easier to use semantic segmentation in new applications by leveraging existing and new pre-trained models. This is a bonus for inclusivity, affordability, and, potentially, environmental impact (as it requires no additional training, which is usually computationally intensive); however, these features also mean that it is easier for bad actors to use the technology.

Because OVDiff does not require further training, it is more versatile, but also, inherits the weaknesses of the components it is built on. For example, it might contain the biases (e.g., gender bias) of its components, in particular Stable Diffusion (Schramowski et al., 2023), which is used for generating support images for any given category/description. Thus it should not be exposed without further filtering and detection of, e.g., NSFW material in the sampled support set. Finally, OVDiff is also bound by the licenses of its components.

## B    OVDIFF: FURTHER DETAILS

In this section, we provide additional details concerning the implementation of OVDiff. We begin with a brief overview of the attention mechanism and diffusion models central to extracting features and sampling images. We review different feature extractors used. We specify the hyperparameter setting for all our experiments and provide an overview of the exchange with ChatGPT used to categorise classes into "thing" and "stuff".

### B.1    PRELIMINARIES

**Attention.** In this work, we make use of pre-trained ViT (Dosovitskiy et al., 2021) networks as feature extractors, which repeatedly apply multi-headed attention layers. In an attention layer, input sequences $X \in \mathbb{R}^{l_x \times d}$ and $Y \in \mathbb{R}^{l_y \times d}$ are linearly project to forms *keys*, *queries*, and *values*: $K = W_k Y$, $Q = W_q X$, $V = W_v X$. In self-attention, $X = Y$. Attention is calculated as $A = \mathrm{softmax}(\frac{1}{\sqrt{d}} Q K^\top)$, and softmax is applied along the sequence dimension $l_y$. The layer outputs an update $Z = X + A \cdot V$. ViTs use multiple heads, replicating the above process in parallel with different projection matrices $W_k, W_q, W_v$. In this work, we consider *queries* and *keys* of attention layers as points where useful features that form meaningful inner-products can be extracted. As we detail later (Appendix B.2), we use the *keys* from attention layers of ViT feature extractors (DINO/MAE/CLIP), concatenating multiple heads if present.

**Text-to-image diffusion models.** Diffusion models are a class of generative models that form samples starting with noise and gradually denoising it. We focus on latent diffusion models (Rombach et al., 2022) which operate in the latent space of an image VAE (Kingma & Welling, 2014) forming powerful conditional image generators. During training, an image is encoded into VAE latent space forming a latent vector $z_0$. A noise is injected forming a sample $z_\tau \sim \mathcal{N}(z_\tau; \sqrt{1 - \alpha_\tau} z_0, \alpha_\tau I)$ for timestep $\tau \in \{1 \ldots T\}$, where $\alpha_\tau$ are variance values that define a noise schedule such that the resulting $z_T$ is approximately unit normal. A conditional UNet (Ronneberger et al., 2015), $\epsilon_\theta(z_t, t, c)$, is trained to predict the injected noise, minimising the mean squared error $\mathbb{E}_t \left( \alpha_t \| \epsilon_\theta(z_t, t, c) - z_0 \|_2 \right)$ for some caption $c$ and additional constants $a_t$. The network forms new samples by reversing the noise-injecting chain. Starting from $\hat{z}_T \sim \mathcal{N}(\hat{z}_T; 0, I)$, one iterates $\hat{z}_{t-1} = \frac{1}{\sqrt{1-\alpha_t}}(\hat{z}_t + \alpha_t \epsilon_\theta(\hat{z}_t, t, c)) + \sqrt{\alpha_t} \hat{z}_t$ until $\hat{z}_0$ is formed and decoded into image space using the VAE decoder. The conditional UNet uses cross-attention layers between image patches and language (CLIP) embeddings to condition on text $c$ and achieve text-to-image generation.

## B.2 Feature extractors

OVDiff is buildable on top of any pre-trained feature extractor. In our experiments, we have considered several networks as feature extractors with various self-supervised training regimes:

- **DINO** (Caron et al., 2021) is a self-supervised method that trains networks by exploring alignment between multiple views using an exponential moving average teacher network. We use the ViT-B/8 model pre-trained on ImageNet[1] and extract features from the *keys* of the last attention layer.

- **MAE** (He et al., 2017) is a self-supervised method that uses masked image inpainting as a learning objective, where a portion of image patches are dropped and the network seeks to reconstruct the full input. We use the ViT-L/16 model pre-trained on ImageNet at a resolution of 448 (Hu et al., 2022).[2] The *keys* of the last layer of the *encoder* network are used. No masking is performed.

- **CLIP** (Radford et al., 2021) is trained using image-text pairs on an internal dataset WIT-400M. We use ViT-B/16 model[3]. We consider two locations to obtain dense features: *keys* from a self-attention layer of the image encoder and *tokens* which are the outputs of transformer layers. We find that *keys* of the second-to-last layer give better performance.

- We also consider **Stable Diffusion**[4] (v1.5) itself as a feature extractor. To that end, we use the *queries* from the cross-attention layers in the UNet denoiser, which correspond to the image modality. Its UNet is organised into 3 downsampling blocks, a middle block, and 3 upsampling blocks. We observe that the middle layers have the most semantic content, so we consider the middle block, 1st and 2nd upsampling blocks and aggregate features from all three cross-attention layers in each block. As the features are quite low in resolution, we include the first downsampling cross-attention layer and the last upsampling cross-attention layer as well. The feature maps are bilinearly upsampled to resolution $64 \times 64$ and concatenated. A noise appropriate for $\tau = 200$ timesteps is added to the input. For feature extraction, we run SD in *unconditional* mode, supplying an empty string for text caption.

## B.3 Datasets

We evaluate on validation splits of PASCAL VOC (VOC), Pascal Context (Context) and COCO-Object (Object) datasets. PASCAL VOC (Everingham et al., 2010; 2012) has 21 classes: 20 foreground plus a background class. For Pascal Context (Mottaghi et al., 2014), we use the common variant with 59 foreground classes and 1 background class. It contains both "things" and "stuff" classes. The COCO-Object is a variant of COCO-Stuff Caesar et al. (2018) with 80 "thing" classes and one class for the background. Textual class names are used as natural language specification of names. We renamed or specified certain class names to fix errors (*e.g.* pottedplant → potted plant), resolve ambiguity better (*e.g.* mouse → computer mouse) or change to more common spelling/word (*e.g.* aeroplane → airplane), resulting in 14 fixes. We experiment and measure an impact of this in Appendix C.1 for our and prior work.

## B.4 Comparative baselines

We briefly review the prior work in Table 1. Most prior work (Liu et al., 2022; Cha et al., 2022; Xu et al., 2022a; Ren et al., 2023; Luo et al., 2022; Xu et al., 2023b) trains image and text encoders on large image-text datasets with a contrastive loss. The methods mainly differ in their architecture and use of grouping mechanisms to ground image-level text on regions. ViL-Seg (Liu et al., 2022) uses online clustering, GroupViT (Xu et al., 2022a) and ViewCo (Ren et al., 2023) employ group tokens. OVSegmentor (Xu et al., 2023b) uses slot-attention and SegCLIP Luo et al. (2022) a grouping mechanism with learnable centers. CLIPPy (Ranasinghe et al., 2022), TCL (Cha et al., 2022), and MaskCLIP (Zhou et al., 2022) predict classes for each image patch: Ranasinghe et al. (2022) use max-pooling aggregation, Cha et al. (2022) self-masking, and Zhou et al. (2022) modify CLIP

---

[1]Model and code available at https://github.com/facebookresearch/dino.

[2]Model and code from https://github.com/facebookresearch/long_seq_mae.

[3]Model and code from https://github.com/openai/CLIP.

[4]We use implementation from https://github.com/huggingface/diffusers.

for dense predictions. To assign a background label (Liu et al., 2022; Cha et al., 2022; Xu et al., 2022a; Ren et al., 2023; Luo et al., 2022) use thresholding while Ranasinghe et al. (2022) uses dataset-specific prompts. ReCO (Shin et al., 2022b) is closer in spirit to our approach as it uses a support set for each prompt; this set, however, is CLIP-retrieved from curated image collections, which may not be applicable for any category in-the-wild.

We also note that prior work builds on top of similar pre-trained components such as CLIP in (Shin et al., 2022b; Zhou et al., 2022; Cha et al., 2022; Luo et al., 2022), DINO + T5/RoBERTa in (Ranasinghe et al., 2022; Xu et al., 2023b). We additionally make use of StableDiffusion, which is trained on a larger dataset (3B, compared to 400M of CLIP). OVDiff is, however, fundamentally different to all prior work, as (a) it generates a support set of synthetic images given a class description, and (b) it does not rely on additional training data and further training for learning to segment.

### B.5 Hyperparameters

OVDiff has relatively few hyperparameters and we use the same set in all experiments. Unless otherwise specified, $N = 32$ images are sampled using classifier-free guidance scale (Ho & Salimans) of 8.0 and 30 denoising steps. We employ `DPM-Solver` scheduler (Lu et al., 2022). When sampling images for the support sets we also use a negative prompt "*text, low quality, blurry, cartoon, meme, low resolution, bad, poor, faded*". If/when CutLER fails to extract any components in a sampled image, a fallback of $M_n^{\text{fb}} = A_n > 0.5$ and $M_n^{\text{bg}} = A_n < 0.2$ is used instead. During inference we set $\eta = 10$, which results in 1024 text prompts processed in parallel, a choice made mainly to due computational constraints. We set the thresholds for the "stuff" filter between background prototypes for "things" classes and the foreground of "stuff" at 0.85 for all feature extractors. When sampling, a seed is set for each category individually to aid reproducibility. With our unoptimized implementation, we measure around $154 \pm 10$s to calculate prototypes for a single category, or $78 \pm 4$s without clustering.

### B.6 Interaction with ChatGPT

We interact with ChatGPT to categorise classes into "stuff" and "things" for stuff filter component. Due to input limits, the categories are processed in blocks. Specifically, we input "*In semantic segmentation, there are "stuff" or "thing" classes. Please indicate whether the following class prompts should be considered "stuff" or "things":*". We show the output in Table 4. Note there are several errors in the response, *e.g.* `glass`, `blanket`, and `trade name` are actually instances of tableware, bedding and signage, respectively, so should more appropriately be treated as "things". Similarly, `land` and `sand` might be more appropriately handled as "stuff", same as `snow` and `ground`. Despite this, We find ChatGPT contains sufficient knowledge when prompted with "in semantic segmentation". We have estimated the accuracy of ChatGPT in thing/stuff classification using the categories of COCO-Stuff, which are defined as 80 "things" and 91 "stuff" categories. ChatGPT achieves an accuracy rate of 88.9% in this case.

## C Additional experiments

In this section, we provide additional experimental results of OVDiff.

### C.1 Additional Comparisons

**Category filter.** To ensure that the category pre-filtering does not give our approach an unfair advantage, we augment two methods (TCL (Cha et al., 2022) and OVSegmentor (Xu et al., 2023b), which are the closest baselines with code and checkpoints available) with our category pre-filtering. We evaluate on the Pascal VOC dataset (where the category filter shows a significant impact, see Table 3) and report the results in Table 5. We observe that TCL improves by 0.6, while the performance of OVSegmentor drops by 0.1. On the contrary, our method benefits substantially from this component, but it still shows stronger performance without the filter than baselines with.

**CutLER (Wang et al., 2023) baseline.** We also further investigate the use of CutLER to obtain segmentation masks. In Table 6, we devise a baseline where CutLER-predicted masks are used to average the CLIP image encoder's final spatial tokens after projection. Averaged tokens are compared

Table 4: **Response from interaction with ChatGPT.** We used ChatGPT model to automatically categorise classes in "stuff" or "things".

| | | | | | |
|---|---|---|---|---|---|
| airplane: | thing | window: | thing | awning: | thing |
| bag: | thing | wood: | stuff | streetlight: | thing |
| bed: | thing | windowpane: | thing | booth: | thing |
| bedclothes: | stuff | earth: | thing | television receiver: | thing |
| bench: | thing | painting: | thing | dirt track: | thing |
| bicycle: | thing | shelf: | thing | apparel: | thing |
| bird: | thing | house: | thing | pole: | thing |
| boat: | thing | sea: | thing | land: | thing |
| book: | thing | mirror: | thing | bannister: | thing |
| bottle: | thing | rug: | thing | escalator: | thing |
| building: | thing | field: | thing | ottoman: | thing |
| bus: | thing | armchair: | thing | buffet: | thing |
| cabinet: | thing | seat: | thing | poster: | thing |
| car: | thing | desk: | thing | stage: | thing |
| cat: | thing | wardrobe: | thing | van: | thing |
| ceiling: | stuff | lamp: | thing | ship: | thing |
| chair: | thing | bathtub: | thing | fountain: | thing |
| cloth: | stuff | railing: | thing | conveyer belt: | thing |
| computer: | thing | cushion: | thing | canopy: | thing |
| cow: | thing | base: | thing | washer: | thing |
| cup: | thing | box: | thing | plaything: | thing |
| curtain: | stuff | column: | thing | swimming pool: | thing |
| dog: | thing | signboard: | thing | stool: | thing |
| door: | thing | chest of drawers: | thing | barrel: | thing |
| fence: | stuff | counter: | thing | basket: | thing |
| floor: | stuff | sand: | thing | waterfall: | thing |
| flower: | thing | sink: | thing | tent: | thing |
| food: | thing | skyscraper: | thing | minibike: | thing |
| grass: | stuff | fireplace: | thing | cradle: | thing |
| ground: | stuff | refrigerator: | thing | oven: | thing |
| horse: | thing | grandstand: | thing | ball: | thing |
| keyboard: | thing | path: | thing | step: | stuff |
| light: | thing | stairs: | thing | tank: | thing |
| motorbike: | thing | runway: | thing | trade name: | stuff |
| mountain: | stuff | case: | thing | microwave: | thing |
| mouse: | thing | pool table: | thing | pot: | thing |
| person: | thing | pillow: | thing | animal: | thing |
| plate: | thing | screen door: | thing | lake: | stuff |
| platform: | stuff | stairway: | thing | dishwasher: | thing |
| plant: | thing | river: | thing | screen: | thing |
| road: | stuff | bridge: | thing | blanket: | stuff |
| rock: | stuff | bookcase: | thing | sculpture: | thing |
| sheep: | thing | blind: | thing | hood: | thing |
| shelves: | thing | coffee table: | thing | sconce: | thing |
| sidewalk: | stuff | toilet: | thing | vase: | thing |
| sign: | thing | hill: | thing | traffic light: | thing |
| sky: | stuff | countertop: | thing | tray: | stuff |
| snow: | stuff | stove: | thing | ashcan: | thing |
| sofa: | thing | palm: | thing | fan: | thing |
| table: | thing | kitchen island: | thing | pier: | thing |
| track: | stuff | swivel chair: | thing | crt screen: | thing |
| train: | thing | bar: | thing | bulletin board: | thing |
| tree: | thing | arcade machine: | thing | shower: | thing |
| truck: | thing | hovel: | thing | radiator: | thing |
| monitor: | thing | towel: | thing | glass: | stuff |
| wall: | stuff | tower: | thing | clock: | thing |
| water: | stuff | chandelier: | thing | flag: | thing |

Table 5: Use of category filter component. OVDiff without category filter outperforms prior work with cat. filter.

| Model | Category filter | |
|---|---|---|
| | ✗ | ✓ |
| OVSegmentor | 53.8 | 53.7 |
| TCL | 51.2 | 51.8 |
| TCL (+PAMR) | 55.0 | 56.0 |
| OVDiff | **56.2** | **66.4** |

Table 6: Application of CutLER. Prior work does not benefit from using CutLER during inference, while OVDiff shows strong results without it.

| Model | CutLER | VOC | Context | Object |
|---|---|---|---|---|
| CLIP | ✓ | 33.0 | 11.6 | 11.1 |
| OVSegmentor | | 53.8 | 20.4 | 25.1 |
| OVSegmentor | ✓ | 38.7 | 14.4 | 16.8 |
| TCL | | 51.2 | 24.3 | 30.4 |
| TCL | ✓ | 43.1 | 20.5 | 22.7 |
| OVDiff | | 62.8 | 28.6 | 34.9 |
| OVDiff | ✓ | **66.3 ± 0.2** | **29.7 ± 0.3** | **34.6 ± 0.3** |

Table 7: Using corrected prompts. We consider if corrected class names benefit prior work. We observe negligible to no effect.

| Model | Correction | VOC | Context | Object |
|---|---|---|---|---|
| OVSegmentor | | 53.8 | 20.4 | 25.1 |
| OVSegmentor | ✓ | 53.9 | 20.4 | 25.1 |
| TCL | | 51.2 | 24.3 | 30.4 |
| TCL | ✓ | 50.6 | 24.3 | 30.4 |
| OVDiff | | 66.1 | 29.5 | 34.9 |
| OVDiff | ✓ | **66.3 ± 0.2** | **29.7 ± 0.3** | **34.6 ± 0.3** |

Table 8: Choice of $K$ for number of centroids.

| K | VOC | Context |
|---|---|---|
| 8 | 63.8 | 29.2 |
| 16 | 64.0 | 29.3 |
| 32 | 64.4 | 29.4 |
| 64 | 64.3 | 28.0 |

with CLIP text embeddings to assign a class. While relying on pre-trained components (like ours), this avoids support set generation. In the same table, we also consider whether the objectness prior provided by CutLER could be beneficial to other methods as well. We consider a version of TCL (Cha et al., 2022) and OVSegmentor (Xu et al., 2023b) which we augment with CutLER. That is, after methods assign class probabilities to each pixel/patch, a majority voting for class is performed in every region predicted by CutLER. This combines CutLER's understanding of objects and their boundaries, aspects where prior methods struggle, with open-vocabulary segmentation. However, we observe that this negatively impacts the performance of these methods, which we attribute to only limited performance of CutLER in complex scenes present in the datasets. Finally, we also include a version of OVDiff that does not rely on CutLER for mask extractions, instead using thresholded masks. We observe that such version of our method also has strong performance, showing that CutLER is helpful but not a critical component and OVDiff performs strongly without it as well.

**Class prompts.** We additionally consider whether corrections introduced to class prompts might have similarly provided additional benefits to our approach. To that end, we also evaluate TCL and OVSegmenter (methods that do not rely on additional prompt curation) with our corrected prompts and consider a version of our method without such corrections in Table 7. We observe only marginal to no impact to the performance.

## C.2 Additional ablations

**Prototype combinations.** In Table 9, we consider the three different types of prototypes described in Section 3 and test their performance individually and in various combinations. We find that the "part" prototypes obtained by $K$-means clustering show strong performance when considered individually on VOC. Instance prototypes show strong individual performance on Context, as well as in combination with the average category prototype. The combination of all three types shows the strongest results across the two datasets, which is what we adopt in our main set of experiments.

We also consider the treatment of prototypes under the stuff filter. We investigate the impact of not excluding background prototypes for "stuff" classes. In this setting, we measure 29.1 on Context, which is a slight reduction in performance. We also investigate the benefit of categorisation into "things" and "stuff" used in the stuff filter component. We instead filter all background prototypes using all foreground prototypes. In this configuration, we measure 27.6 on Context. Both configurations

Table 9: Ablation of various configurations for prototypes. We consider average $\bar{P}$, instance $P_n$, and part $P_k$ prototypes individually and in various combinations on VOC and Context datasets. Combination of all three types of prototypes shows strongest results.

| $\bar{P}$ | $P_n$ | $P_k$ | VOC | Context |
|---|---|---|---|---|
| ✓ | ✓ | ✓ | 64.4 | 29.4 |
| ✓ |  | ✓ | 61.7 | 29.3 |
| ✓ | ✓ |  | 63.5 | 29.4 |
|  | ✓ | ✓ | 62.5 | 28.4 |
|  |  | ✓ | 63.7 | 28.8 |
|  | ✓ |  | 60.0 | 29.0 |
| ✓ |  |  | 62.5 | 28.4 |

Table 10: Ablation of different SD feature configurations. Removing first and last cross attention *layers*, mid, 1st and 2nd upsampling *blocks* (all layers in the block) has a negative effect.

| 1st layer | Mid block | Up-1 block | Up-2 block | Last layer | Context |
|---|---|---|---|---|---|
| ✓ | ✓ | ✓ | ✓ | ✓ | 29.4 |
|  | ✓ | ✓ | ✓ | ✓ | 29.4 |
| ✓ |  | ✓ | ✓ | ✓ | 29.2 |
| ✓ | ✓ |  | ✓ | ✓ | 27.3 |
| ✓ | ✓ | ✓ |  | ✓ | 28.9 |
| ✓ | ✓ | ✓ | ✓ |  | 29.3 |

Table 11: Comparison with methods when background is *not* considered. We compare OVDiff with prior work on VOC-20, Context-59 and ADE datasets in a setting that considers only the foreground pixels (decided by ground truth). Our method shows comparable performance to prior works despite only relying on pretrained feature extractors. Our results are an average of 5 seeds $\pm\sigma$. * result from (Cha et al., 2022).

| Method | VOC-20 | Context-59 | ADE-150 |
|---|---|---|---|
| GroupViT* | 79.7 | 23.4 | 9.2 |
| MaskCLIP* | 74.9 | 26.4 | 9.8 |
| ReCo* | 57.5 | 22.3 | 11.2 |
| PACL | 72.3 | **50.1** | **31.4** |
| TCL | 77.5 | 30.3 | 14.9 |
| **OVDiff** | **80.2 ± 0.6** | 33.0 ± 0.2 | 14.1 ± 0.2 |

show a reduction from 29.4, measuring using the stuff filter with categorisation in "stuff" and "things", as used in our main experiments.

$K$ **- number of clusters.** In Table 8, we investigate the sensitivity of the method to the choice of $K$ for the number of "part" prototypes extracted using $K$-means clustering. Although our setting $K = 32$ obtains slightly better results on Context and VOC, other values result in comparable segmentation performance suggesting that OVDiff is not sensitive to the choice of $K$ and a range of values are viable.

**SD features.** When using Stable Diffusion as a feature extractor, we consider various combinations of layers/blocks in the UNet architecture. We follow the nomenclature used in the Stable Diffusion implementation where consecutive layers of Unet are organised into *blocks*. There are 3 down-sampling blocks with 2 cross-attention layers each, a mid-block with a single cross-attention, and 3 up-sampling blocks with 3 cross-attention layers each. We report our findings in Table 10. Including the first and last cross-attention layers in the feature extraction process has a small positive impact on segmentation performance, which we attribute to the high feature resolution. We also consider excluding features from the middle block of the network due to small $8 \times 8$ resolution but observe a small negative impact on performance on Context dataset and substantial decrease on VOC. We also investigate whether including the first (Up-1) and the second upsampling (Up-2) blocks are necessary. Without them, the performance drops the most out of the configurations considered. Thus, we use a concatenation of features from the middle, first and second upsampling blocks and the first and last layers in our main experiments.

## C.3 EVALUATION WITHOUT BACKGROUND

One of the notable advantages of our approach is the ability to represent background regions via (negative) prototypes, leading to improved segmentation performance. Nevertheless, we hereby also evaluate our method under a different evaluation protocol, adopted in prior work, which excludes the

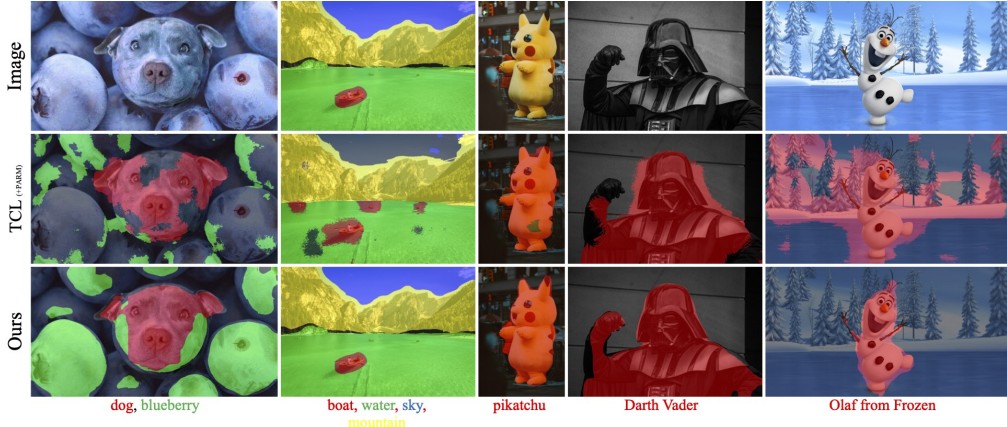

Figure 6: Qualitative comparison on in-the-wild images. OVDiff performs significantly better than prior state-of-the-art, TCL, on a confusing composite (photoshopped) image, a scenery photo, and realistic and cartoon images containing popular characters.

*background* class from the evaluation. We note that prior work often requires additional considerations to handle background, such as thresholding. In this setting, however, the background class is *not* predicted, and the set of categories, thus, must be exhaustive. As in practice this is not the case, and datasets contain unlabelled pixels (or simply a background label), such image areas are removed from consideration. Consequently, less emphasis is placed on object boundaries in this setting. We test our method on three datasets: PascalVOC without background termed VOC-20, Pascal Context without background termed Context-59, and ADE20k (Zhou et al., 2017) which contains 150 foreground classes. As in this setting the background prediction is invalid, we do not consider negative prototypes. This setting tests the ability of various methods to discriminate between different classes, which for OVDiff is inherent to the choice of feature extractors. Despite this, our method shows competitive performance. There exists a notable gap between PACL and other works, including ours, on Context-59 and ADE-150. In the case of OVDiff, we attribute this to the limited resolution of our feature extractors, especially on ADE-150 where a variety of tiny objects is present. PACL, on the other hand, proposes a method to increase the resolution of their trained network 4 times during inference.

## C.4 QUALITATIVE RESULTS

We include additional qualitative results from the benchmark datasets in Fig. 7. Our method achieves high-quality segmentation across all examples, without any post-processing or refinement steps. Finally, in Fig. 8, we show examples of support images sampled for some thing, and stuff categories.

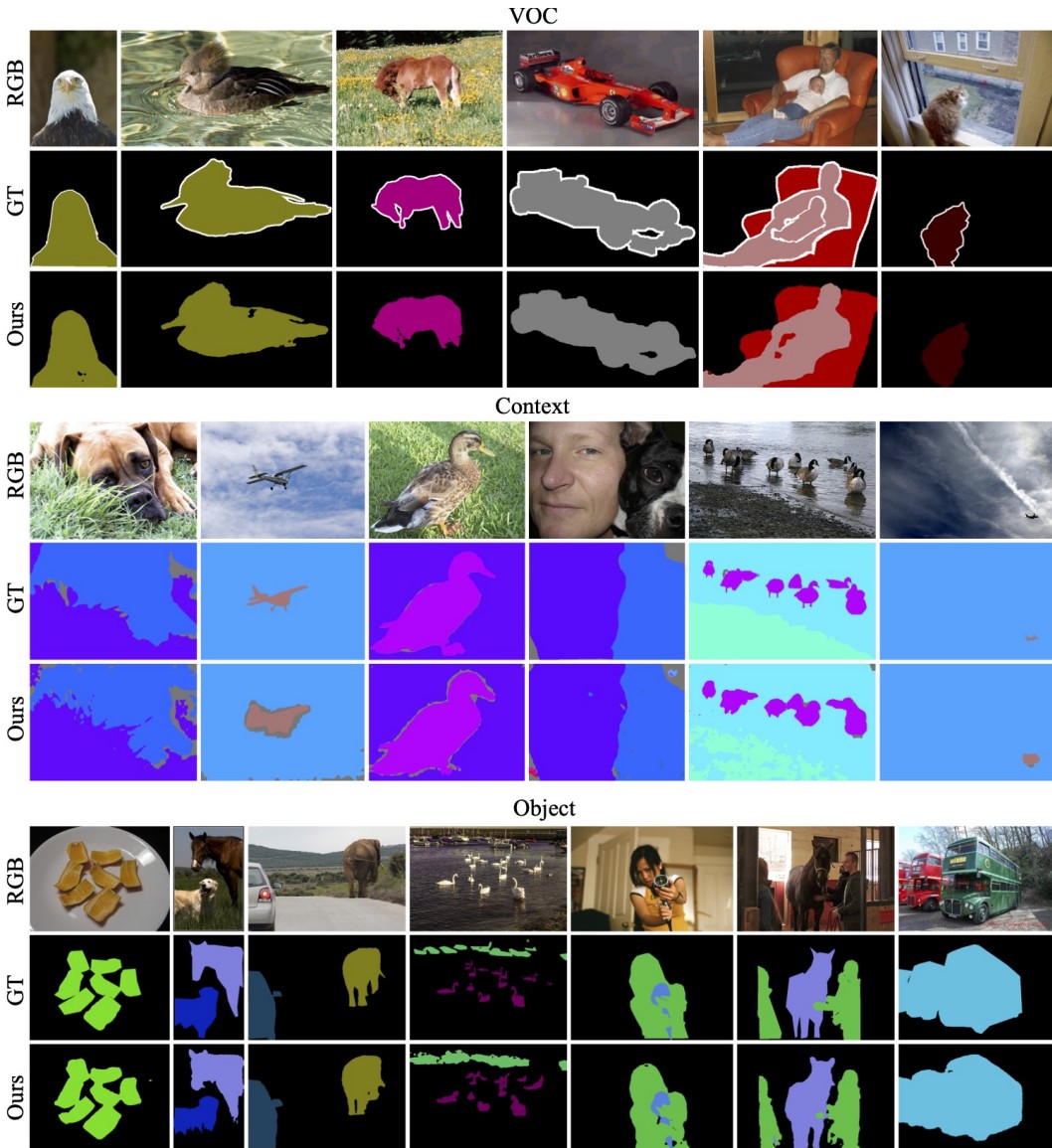

Figure 7: Additional qualitative results. Images from Pascal VOC (top), Pascal Context (middle), and COCO Object (bottom).

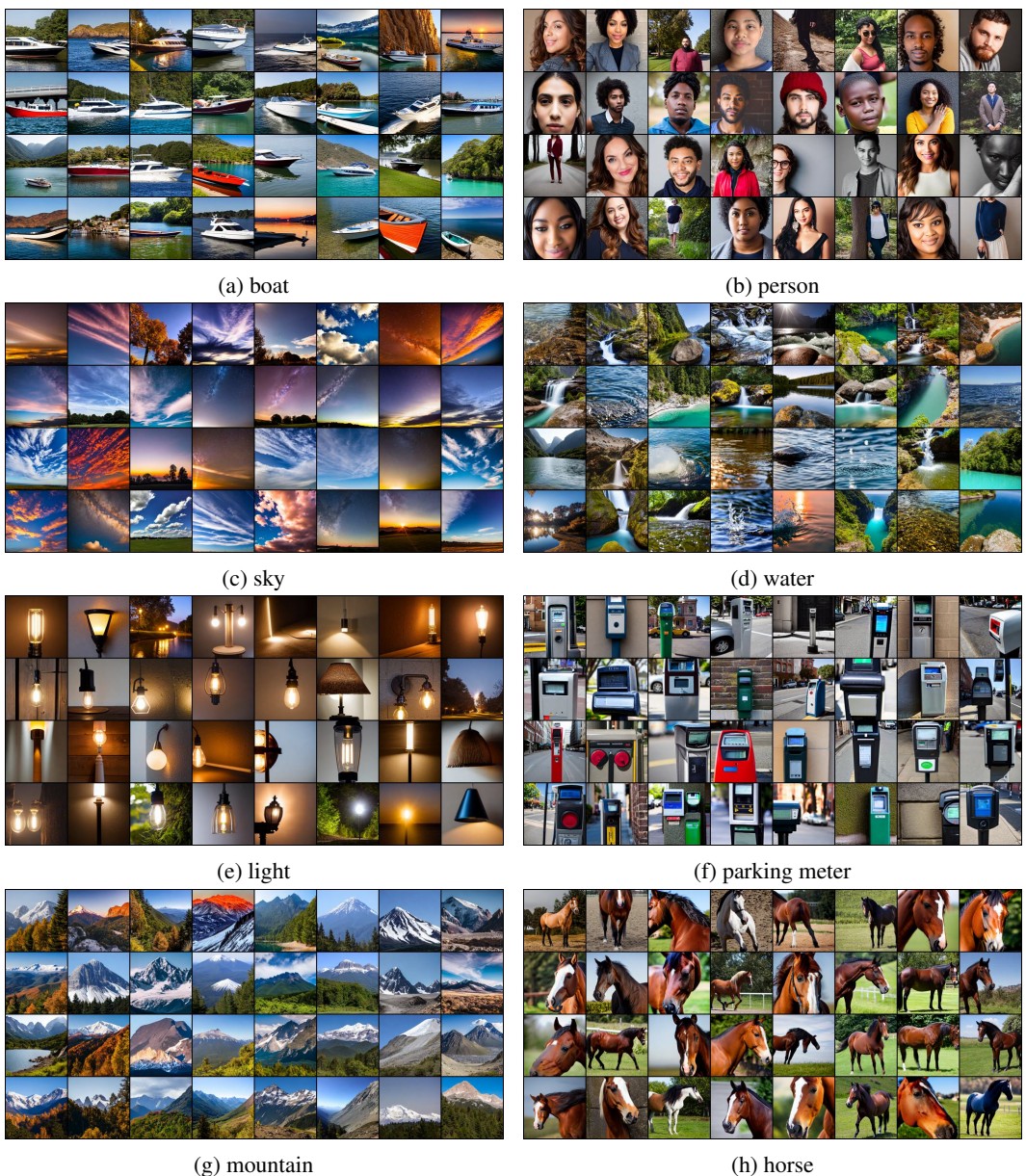

(a) boat

(b) person

(c) sky

(d) water

(e) light

(f) parking meter

(g) mountain

(h) horse

Figure 8: Images sampled for a support set of some categories.

