# OpenReview forum: "Diffusion Models for Open-Vocabulary Segmentation"
_ICLR.cc/2024/Conference — ICLR 2024 Conference Withdrawn Submission_

### Official Review · Reviewer_FmYQ · 2023-10-21

**Soundness:** 3 good
**Presentation:** 3 good
**Contribution:** 3 good
**Rating:** 6
**Confidence:** 5

**Summary:**

This paper introduces a novel method for open vocabulary segmentation. Without the need for training,  it leverages diffusion models to generate examples for the categories uses the clip/dino to extract prototypes, and uses the prototypes to make segmetation.

**Strengths:**

1. This work is novel and interesting.  It provides a new idea to tackle open vocabulary segmentation.
2. The motivation is clear writing is easy to understand.
3. Thanks to the generalization ability of SD, CLIP, and DINO models, the proposed methods show a strong generalization ability for "zero-shot" tasks.

**Weaknesses:**

1. The definition of "zero-shot". As the authors use diffusion models to generate images for the potential categories, I suggest the authors not claim "zero-shot". Because using SD to generate images is somehow equivalent to collecting the target images from the internet. The categories are no longer "unseen".  From my perspective, "open-vocabulary" is acceptable but "zero-shot" is not.

2. Burdens of generating a large support set.  Although this work does not require training, generating and storing the support set might be heavy when the category list becomes large.

**Questions:**

See the weaknesses.

---

> ### Author Response · Authors · 2023-11-14
>
> We thank the reviewer for finding our work novel and interesting, with clear writing and strong results.
>
> > The definition of "zero-shot". As the authors use diffusion models to generate images for the potential categories, I suggest the authors not claim "zero-shot". Because using SD to generate images is somehow equivalent to collecting the target images from the internet. The categories are no longer "unseen". From my perspective, "open-vocabulary" is acceptable but "zero-shot" is not.
>
> We agree with the reviewer that the term "zero-shot" has been drifting in its meaning over the years. Originally it was used only for recognition through another modality; since then, papers such as CLIP and prior work to ours have been using it more liberally (often to indicate that the method has not been trained on the dataset it is evaluated on).
>
> For this reason, we did not claim to introduce a zero-shot method anywhere in this paper. We will make this more explicit and include a discussion of the term.
>
> > Burdens of generating a large support set. Although this work does not require training, generating and storing the support set might be heavy when the category list becomes large.
>
> Our raw storage requirement is 0.39MB per class using CLIP, which is feasible even for hundreds or thousands of classes. Note that only the prototype vectors need to be stored if storage is a concern. As inference is performed using nearest-neighbour look-up (cosine distance), a fast NN look-up data structure can be leveraged for this. This would be similar to using CLIP for retrieval: instead of CLIP descriptors for image collections, prototype vectors would be indexed and stored. Features of the input image would be used to query the data structure, returning the best match.
>
> It takes around $154\pm10$s to sample prototypes for a single category (L664-665). It would require around 0.18 GPU days for 100 classes, which we believe is reasonable. Note, we have not optimised to lower this number in this study, and further gains are very likely, e.g., clustering is currently done on the CPU. Similarly, sampling times could be cut down significantly by considering better SD schedulers such as the recent LCM (Lou et al., 2023), which offers 5-6x speedup.
>
> ---
> (Lou et al., 2023) Latent Consistency Models: Synthesizing High-Resolution Images with Few-Step Inference, 2023

---

### Official Review · Reviewer_zXW9 · 2023-11-01

**Soundness:** 3 good
**Presentation:** 3 good
**Contribution:** 2 fair
**Rating:** 5
**Confidence:** 3

**Summary:**

The paper describes OVDiff, a model that uses text-to-image diffusion models for open-vocabulary segmentation.

The basic approach is to:
a) use text queries with a text-to-image model to produce sample images that constitute a support set.
b) unsupervised instance segmentation is used (e.g., CutLER) along with cross-attention maps of the diffusion model to distinguish between foreground and background in the support set images.
c) from the support set, prototypes are learned for the class, instance and parts. Both the object and the background are used for positive and negative prototypes.
d) finally, a segmentation map is obtained by comparing dense image feature to prototypes using cosine similarity.

Experiments are performed with several image encoders; DINO, MAE, SD (stable diffusion) and CLIP, and on several datasets; PASCAL VOC, Pascal Context, and COCO-object. Several ablations are performed, e.g., combining image features outperforms any individual feature.

**Strengths:**

Overall the paper is well written and clearly lays out the approach and the experiments.

The proposed method relies on off-the-shelf pretrained components, and it is relatively straightforward.

Experiments explore a few interesting ablations, e.g., the contribution from the background components, the distinction between stuff and things, etc.

**Weaknesses:**

Several related missing references:
Learning to Detect and Segment for Open Vocabulary Object Detection, T. Wang, N. Li, CVPR 2023
Semantic-SAM: Segment and Recognize Anything at Any Granularity, Feng Li et al., https://arxiv.org/pdf/2307.04767.pdf

The paper does not describe results on LVIS, commonly used for open set segmentation.

In distinguishing between stuff and things, the authors describe asking ChatGPT. It was unclear to me whether this was necessary for the paper as it was a relatively small number of classes and the results contained some errors. It's possible that better prompting could have produced more accurate results.

The results shown in Fig. 5 show a few issues w/ OVDiff; e.g., small false positive "corgi" patches, issues w/ the donut image, a small false positive patch of "bus".

Editing Comments
p. 6: As the approach does note require --> As the approach does not require
p. 8: Though sometimes region --> Though sometimes the region?
p. 8: not fully align with whole --> not fully align with the whole?

**Questions:**

a) Please clarify the overall novelty and contribution of the paper.
b) In the text-to-image methodology, it seems that only single-class prompts are used (e.g., "a good picture of a cat" or "a good picture of a dog") rather than more complex queries that could provide more shape information when segmenting. Does this limitation impact performance?
c) Comment on whether it would be useful to benchmark on LVIS?

---

> ### Author Response · Authors · 2023-11-14
>
> We thank the Reviewer for recognizing our interesting ablation experiments and clear writing.
>
> > Several related missing references: Learning to Detect and Segment for Open Vocabulary Object Detection, T. Wang, N. Li, CVPR 2023 Semantic-SAM: Segment and Recognize Anything at Any Granularity, Feng Li et al., https://arxiv.org/pdf/2307.04767.pdf
>
> Thank you for the references. We will include them in the RW section.
> In brief, Wang and Li introduce a CondHead module to improve other supervised open-vocabulary detection and segmentation methods. Feng Li et al. propose a method to train an open-set segmenter by decoupling object and part classification and training on several segmentation tasks.
>
> > In distinguishing between stuff and things, the authors describe asking ChatGPT. It was unclear to me whether this was necessary for the paper as it was a relatively small number of classes and the results contained some errors. It's possible that better prompting could have produced more accurate results.
>
> The use of ChatGPT is not strictly necessary in practice as one would likely know whether a prompt describes a "thing" or a "stuff". However, we wanted to present a more complete method that does not rely on such additional input. We have experimented with providing "oracle" answers "stuff" or "thing" classification. We measure 29.6 mIoU on the Context dataset ($29.7\pm0.3$ with ChatGPT), showing that the minor errors do not impact the results.
>
> We agree that it is entirely possible that different prompting could produce better results. It is an interesting question for a future direction whether this can be improved or exchanged for an alternative approach.
>
> > The results shown in Fig. 5 show a few issues w/ OVDiff; e.g., small false positive "corgi" patches, issues w/ the donut image, a small false positive patch of "bus".
>
> In the caption, we will more clearly indicate that we also aim to show minor failures in these challenging in-the-wild images. We decided to show samples representative of the method's performance over cherry-picking perfect results. Note that compared to the best available prior work, however, our method makes fewer mistakes: capturing whole objects and not getting confused by similar color attributes.

---

> ### Author Response · Authors · 2023-11-14
>
> > Editing Comments:
>
> Thank you. We shall incorporate these!
>
> > a) Please clarify the overall novelty and contribution of the paper.
>
> We introduce a novel framework, OVDiff, that diverges significantly from previous methods. With OVDiff, it becomes possible to surpass the state of the art in open vocabulary unsupervised segmentation by purely relying on pre-trained foundation models without further training. The idea of creating prototypes from generated data for OV segmentation is new, and all associated components (such as bg prototypes) are original (e.g. L187, L190-207, L214, L224) -- unless otherwise stated. We consider the following our core contributions:
>  - A method for using pre-trained diffusion models for OV semantic segmentation without manual annotations or further training.
>  - A principled way to handle backgrounds by forming prototypes from contextual priors built into text-to-image generative models.
>  - A set of additional techniques for further improving performance, such as multiple prototypes, category filtering and "stuff" filtering.
>
> > b) In the text-to-image methodology, it seems that only single-class prompts are used (e.g., "a good picture of a cat" or "a good picture of a dog") rather than more complex queries that could provide more shape information when segmenting. Does this limitation impact performance?
>
> Thank you for this suggestion! We considered a single prompt the simplest solution broadly applicable to virtually any natural language specification of a target class, not just limited to, e.g. class prompts present in standard benchmarks. While prior work adopts prompt expansion by considering a list of synonyms and subcategories, it is not entirely clear how such a strategy could be systematically performed for any in-the-wild prompts, such as a “chocolate glazed donut”. It is also not clear how a more complex query could be constructed. One approach adopted in prior works is considering _multiple_ similar queries via synonyms.
>
> Based on the reviewer’s suggestion, we experimented with a list of synonyms and subclasses, as employed in (Ranasinghe et al., 2022). However, we did not observe a significant impact, measuring 66.4 mIoU on VOC (c.f. $66.3\pm0.2$). Curating such lists automatically is an interesting future scaling direction.
>
> > The paper does not describe results on LVIS, commonly used for open set segmentation.
> > c) Comment on whether it would be useful to benchmark on LVIS?
>
> LVIS is a popular benchmark for open-set segmentation. However, LVIS is an _instance_ segmentation dataset that is popular for _supervised_ methods. As we consider the semantic segmentation task and do not use mask supervision, we adopted datasets and benchmarks used in prior comparable work to offer a fair comparison for this task. To the best of our knowledge no other comparable method has been evaluated on the LVIS benchmark. We will attempt to compute the results for the final version.

---

### Official Review · Reviewer_kQ2p · 2023-11-02

**Soundness:** 3 good
**Presentation:** 3 good
**Contribution:** 3 good
**Rating:** 6
**Confidence:** 3

**Summary:**

This work proposes to leverage the generative text-to-image diffusion models to enhance open-vocabulary segmentation. The proposed method OVDiff synthesizes support image sets from category names and collect the representative prototypes for each category. The segmentation is performed by comparing a target image with the prototypes.

**Strengths:**

* The proposed idea of using text-to-image generated samples as support set  to perform the image-image feature comparison seems novel.
* OVDiff achieves the state-of-the-art on VOC, Context and Object benchmarks.
* OVDiff also exhibits reasonable segmentation on the in-the-wild examples.

**Weaknesses:**

* The background segmentation requires a pre-computation and the use of external module CutLER (Wang et al. 2023).

* It seems requiring a careful curation and parameter control to achieve the accurate foreground/background segmentation and to collect the good representative prototypes.

* As the synthesized images are mostly object-centric, it is not clear whether the method can still work on large images with multiple fine-grained objects.

* When evaluated on the context-59 and ADE-150 datasets with more fine-grained objects, OVDiff performs worse than some of the recent SOTA methods.

* While running speed is not a main benchmark in open-vocabulary segmentation, the proposed pipeline of image synthesis, prototype collection, background computation clustering seems involving quite a bit of computation.

**Questions:**

* Would OVDiff run faster or slower than the SOTA methods?
* Please see Weaknesses section for other questions.

---

> ### Author Response · Authors · 2023-11-14
>
> We thank the reviewer for recognising the novelty of our method and strength of our results.
>
> > The background segmentation requires a pre-computation and the use of external module CutLER (Wang et al. 2023).
>
> We use CutLER to improve the segmentation of the sampled support images. CutLER is _not used_ during inference. Segmentation for the "background" class (more precisely, a category that encompasses everything else not in the vocabulary) is achieved by comparing to background prototypes of all other categories. This leverages the contextual prior of categories captured in the generative model.
>
> Note that while CutLER improves our performance slightly, it is not critical (Table 6. Appendix C). Our experiments in Appendix C show that our method _without CutLER_ performs similarly on Context and Object datasets and has a slight drop in performance on PASCAL VOC. OVDiff _without CutLER_ still achieves state-of-the-art performance.
>
> > It seems requiring a careful curation and parameter control to achieve the accurate foreground/background segmentation and to collect the good representative prototypes.
>
> Our method has very few hyper-parameters and we use the same set across all datasets. We evaluate hyper-parameter sensitivity in Figure 4 and Tables 8, 9 and 10. While the experiments confirm our choice of values, the performance does not vary much when changing hyper-parameters, demonstrating that our method is not particularly sensitive to these choices and a range of values can work.
>
> We also emphasise _no curation_ is performed at all. We are not sure what "careful curation" refers to. The set of 32 images is sampled and processed automatically to build prototypes. Furthermore, we report our results accross 5 different seeds. While there is some variation in the performance as could be expected, the variances are low.
>
> > As the synthesized images are mostly object-centric, it is not clear whether the method can still work on large images with multiple fine-grained objects.
>
> It does! We achieve strong results on COCO-Object (which uses 80 MSCOCO categories and merging of instance masks into semantic ones) and Pascal Context. Both datasets are scene-centric featuring a large number of classes and often many small objects. E.g. in Fig. 7 we segment many small birds and boats in the Object dataset.
>
> OVDiff, as an approach, capitalises on the fact that the sampled images are object-centric. This makes it easier to separate foreground class and contextual backgrounds (with just attention or optionally with CutLER for added improvement) to construct the prototypes. The calculated prototypes then generalise to scene-centric images via feature matching, to achieve state-of-the-art performance. Feature matching is not impacted by the distribution shift as it is carried out locally (between image patches and prototypes).

---

> ### Author Response · Authors · 2023-11-14
>
> > When evaluated on the context-59 and ADE-150 datasets with more fine-grained objects, OVDiff performs worse than some of the recent SOTA methods.
>
> As we discuss in Section C.3 (Appendix), the Context-59 and ADE-150 benchmarks present a different setting as they exclude background and unlabelled pixels from the evaluation. We did not consider this setting our "main" benchmark as the it relies on both having an _exhaustive_ set of classes to model the world and/or _knowing_ which pixels cannot be modelled. We deem that neither of these assumptions would apply to a practical application. Instead, we considered a setting where background also needs to be predicted, to effectively segment regions that do not match any of the classes in a given vocabulary.
>
> However, for completeness and fairness to prior work, we also compare to the setting where the background is excluded and thus none of the methods are actually required to reason about it. We note that although this setting does not enable us to make full use of the proposed components (such as the negative prototypes for background), we still show strong results. Since the PACL implementation is not available, it is difficult to elaborate more on their performance.
>
> Regarding fine-grained object categories, in particular, we show qualitative examples in Figures 5 and 6. We note that several images in those figures do require fine-grained recognition, e.g., the first (cat-poodle-corgi-baby) and fourth (donuts) examples in Fig. 5 and the "dog or blueberry" example in Fig. 6. Our method makes fewer mistakes and returns better boundaries.
>
> > While running speed is not a main benchmark in open-vocabulary segmentation, the proposed pipeline of image synthesis, prototype collection, background computation clustering seems involving quite a bit of computation.
> > Would OVDiff run faster or slower than the SOTA methods?
>
> During inference OVDiff is slower but still comparable with other methods: 0.6s for OVDiff, 0.2s for TCL, and 0.08s for OVSegmentor.
>
> OVDiff is a training-free approach; instead of a training phase it pre-computes a set of prototypes for a number of categories. It takes $154\pm10$s to sample prototypes for a single category; this is required only once per category and can then be reused.
> As the prototypes for the categories are independent, this is can be done offline and/or in parallel as well. Similarly, the set of prototypes can later be expanded with new categories.
>
> Comparing to the training requirements of other methods, this is rather cheap. The time required to compute prototypes even for larger vocabularies (e.g., 0.18 GPU days for 100 categories) is significantly smaller than training a closed-set segmentation network (e.g., Mask-RCNN takes ~10 GPU days) or the training times of other unsupervised open-vocabulary methods: 32 GPU days for GroupVit, 40+ GPU days for PACL, and 8 GPU days for TCL.
>
> Note, that in practice semantic segmentation models are deployed with rarely changing vocabulary. This is because altering vocabulary for each image in principled way would require some knowledge of its contents and might be better modelled by other segmentation tasks like referrent expression segmentation. In that respect, prototype sampling is heavily amortised.

---

### Official Review · Reviewer_PKyW · 2023-11-08

**Soundness:** 2 fair
**Presentation:** 2 fair
**Contribution:** 2 fair
**Rating:** 3
**Confidence:** 4

**Summary:**

This paper present OVDiff, a novel method that leverages the generative properties of text-to-image diffusion models for open-vocabulary segmentation. The proposed method shows good results on PASCAL VOC.

**Strengths:**

1. The proposed method achieve SOTA performance on challenging benchmarks
2. Figures in the paper are clear and easy to follow

**Weaknesses:**

1. The paper mentions the use of diffusion to generate images and extract the corresponding feature prototypes. However, this approach may introduce bias due to the potentially limited diversity of the generated images, leading to biased results. Generating a larger number of images to address this issue would result in a significant increase in time, as creating a single image with diffusion methods is time-consuming, requiring at least 2-3 seconds even when accelerated by methods like Denoising Diffusion Implicit Models (DDIM).

2. The core insight of your study is not immediately clear. Could you succinctly summarize the key findings and the experimental evidence that supports them? The method section could be simplified for better readability; currently, the intertwining of motivation within the methodological steps detracts from a clear understanding.
Furthermore, there is a discrepancy between the subtitles in Figure 1 and the corresponding method section headings, which disrupts the flow for the reader. The methods section itself seems overly intricate, resembling a layered application of preexisting large models and techniques from other studies, which dilutes the novelty of your work.

3. I am concerned about the complexity of the process and would like to know detailed information on the time required to process a single image, including the computational costs needed for the entire procedure from image generation to final segmentation results. Additionally, how does this compare to other methods? A report on the processing times for alternative methods is also necessary for a thorough comparison.

**Questions:**

Please refer to the weaknesses.

---

> ### Author Response · Authors · 2023-11-14
>
> We thank the reviewer for recognising the novelty and strong performance of our approach and the clarity of our figures.
>
> > this approach may introduce bias due to the potentially limited diversity of the generated images, leading to biased results. Generating a larger number of images to address this issue would result in a significant increase in time [...]
>
> The following aspects of our design contribute to mitigating potential biases that arise from limited diversity:
>
> (1) To aid with the potentially limited diversity of the support set, we include diverse types of prototypes in addition to just instance prototypes, namely class-level and part-level (L196-202), which provide alternative means for assessing similarity rather than just comparing instances.
>
> (2) More importantly, to segment a target image, the comparison between the image and the prototypes is done in _feature space_ and not in RGB space. Because the feature extractors considered in this work (e.g., SD, CLIP, DINO) have already learned rich _semantic_ spaces, different instances of the same class should map to similar vectors irrespective of their intra-class appearance variations. This is true for both prototypes and target image features.
>
> We illustrate this further with qualitative examples. We show sampled images for the "airplane" and "car" classes [here](https://imageupload.io/byEY6AvvkLMhEWz). Although most airplane images feature commercial airplanes and most cars are consumer sedans, our method is able to segment a small propeller plane and a formula vehicle in Figure 7, Appendix C.
>
> Finally, we note that increasing the support set size does not lead to notable performance gains. This is supported by Figure 4, discussed in Section 4.3 (L296-301), where we experiment with increasing the size of the support set. We observe that as the support size increases, the segmentation performance changes only very slightly and mostly saturates at 32 samples per class.
>
> > The core insight of your study is not immediately clear. Could you succinctly summarize the key findings and the experimental evidence that supports them?
>
> The key insight is that the proposed method makes it possible to segment images with an open vocabulary by relying purely on pre-trained foundation models without any further training, even though these models are not designed for dense (e.g., segmentation) tasks.
>
> This approach is novel and diverges significantly from previous works that, instead, train models to achieve dense alignment of images and text. The idea of creating prototypes from generated data for open-vocabulary segmentation is new, and all associated components (such as bg prototypes) are original -- unless otherwise stated (e.g. L187, L190-207, L214, L224). We will clarify this in the paper.
>
> We consider the following our core contributions:
>  - A method to use pre-trained diffusion models for the task of OV semantic segmentation without manual annotations or further training.
>  - A principled way to handle backgrounds by forming prototypes from contextual priors built into text-to-image generative models.
>  - A set of additional techniques for further improving performance, such as multiple prototypes, category filtering and "stuff" filtering.
>
> We support this with quantitative evaluations on several widely adopted benchmarks (Table 1), surpassing the state-of-the-art in open-vocabulary unsupervised segmentation. We conduct ablation experiments verifying the necessity of the components employed in our method (Tables 2 & 3).

---

> ### Author Response · Authors · 2023-11-14
>
> >The method section could be simplified for better readability; currently, the intertwining of motivation within the methodological steps detracts from a clear understanding.
>
> We shall update the methods section with a concise component summary before delving into the details to better guide the reader.
>
> > Furthermore, there is a discrepancy between the subtitles in Figure 1 and the corresponding method section headings, which disrupts the flow for the reader.
>
> Thank you for pointing this out. We will ensure the labelling is consistent across the figure and the section.
>
> > The methods section itself seems overly intricate, resembling a layered application of preexisting large models and techniques from other studies, which dilutes the novelty of your work.
>
> Thanks! We will streamline the methods section and move details to the appendix.
>
> > I am concerned about the complexity of the process and would like to know detailed information on the time required to process a single image, including the computational costs needed for the entire procedure from image generation to final segmentation results. Additionally, how does this compare to other methods? A report on the processing times for alternative methods is also necessary for a thorough comparison.
>
> As we report in Appendix B.5 (L664-665), we measure around $154\pm10$s to sample prototypes for a single category. Note that this number can be further improved by better implementation (e.g., we currently cluster on CPU).
>
> Importantly, we also note that the sampling process is only required _once_ per category/prompt as opposed to once per image. This means that it can be parallelised, and the prototype set can be stored and even later expanded. We consider this to be relatively cheap procedure as even for a reasonably large vocabulary of one hundred classes, it would take around 0.18 GPU days. This is quite small compared to the training times of prior work, such as 32 GPU days for GroupVit, 40+ GPU days for PACL and 8 GPU days for TCL.
>
> We measure 0.6s per image during inference which is slower but comparable to 0.2s of TCL and 0.08s of OVSegmentor. We performed measurements using SD on 2080Ti GPU using 21 classes and same resolution/sliding window settings for all methods.

---

### Author Response · Authors · 2023-11-15
**Global Response**

We thank the reviewers for their valuable feedback and for recognising our novelty and strong results (PKyW, kQ2p, FmYQ) and clarity of presentation (PKyW, zXW9, FmYQ). We have addressed the comments and answered the question individually below, and we will work on updating the manuscript in the coming days.